# Study on Dynamic Probability and Quantitative Risk Calculation Method of Domino Accident in Pool Fire in Chemical Storage Tank Area

**DOI:** 10.3390/ijerph192416483

**Published:** 2022-12-08

**Authors:** Mingqing Su, Lijun Wei, Shennan Zhou, Guoliang Yang, Rujun Wang, Yingquan Duo, Sining Chen, Mingliang Sun, Jiahang Li, Xiangbei Kong

**Affiliations:** 1Key Laboratory of Environmental Change and Natural Disaster, Ministry of Education, Beijing Normal University, Beijing 100875, China; 2China Academy of Safety Science and Technology, Beijing 100012, China; 3School of Emergency Management and Safety Engineering, China University of Mining and Technology (Beijing), Beijing 100083, China; 4Key Laboratory of Major Hazard and Chemical Industry Park System Safety, Ministry of Emergency Management of China, Beijing 100012, China; 5State Key Laboratory of Explosion Science and Technology, Beijing Institute of Technology University, Beijing 100081, China

**Keywords:** domino fire accident, Monte Carlo simulation, synergic effect, chemical storage tank, judgment criterion, quantitative risk analysis

## Abstract

The domino event caused by fire is one of the common accidents in hydrocarbon storage tank farms, which further expands the severity and scope of the accident. Due to the different failure sequence of the storage tanks in a domino accident, the radiant heat generated by the failed storage tank to the target tank is different. Based on the influence of this synergistic effect, this study combined the Monte Carlo algorithm and FSEM, and proposed a fast real-time probability calculation method for a fire domino accident in a storage tank area, for the first time. This method uses the Monte Carlo algorithm to simulate all accident scenarios, and obtains the evolution of multiple escalation fire domino accidents under the synergistic effect according to FSEM, and then calculates the real-time failure probability and risk. Based on a comprehensive analysis of the accident propagation path, this method avoids the problem of a large amount of calculation, and is conducive to the rapid and effective analysis of the fire risk in a storage tank area and the formulation of corresponding risk reduction measures. The effectiveness and superiority of the proposed method were proved by a case study.

## 1. Introduction

With the development of chemical industry parks (chemical clusters), the tank area, as an important area for the storage and transportation of chemical raw materials, intermediates and finished products, has shown a trend in recent years of having large-scale tanks and diversified storage types [1]. As the process pipelines in the tank area are various and complex, and most of the storage media are hazardous chemicals with flammable and explosive characteristics, once leakage occurs, fire and explosion accidents are likely to occur [2,3,4]. The thermal radiation [5,6], explosion overpressure [7,8] and explosion debris [9] generated by these initial events, as well as the interaction between them, often lead to cascading domino accidents, causing serious casualties and property losses [10]. Research also shows that domino accidents are more likely to occur in the storage process than in production, transportation and other links. Moreover, domino accidents caused by fire accounted for the highest proportion, at least 32%, of accident data [11,12,13]. Therefore, the domino accident caused by a tank fire requires research focus.

The reasons for a domino accident caused by an initial tank fire differ. When the oil in a tank contains water, if the oil in the upper layer burns, the water in the lower layer, that is lighter than the oil, is difficult to volatilize, thus, forming expanding gas that causes the liquid surface to boil, which causes the tank that initially caught fire to boil over. The spilled ignited oil droplets cause adjacent tanks to burn [14,15,16]. In the combustion process of the storage tank, once leakage occurs, the oil spreads and diffuses on the ground, so that the combustion area gradually expands and flowing fire forms, which leads to fire in the adjacent storage tanks [17,18]. In addition, the initial fire storage tank transfers heat to the adjacent storage tank in the form of thermal radiation during the combustion process, resulting in the destruction of its structural integrity, and, thus, causing combustion [19,20]. In fact, the domino accident caused by tank fire is often the result of the interaction of several scenarios [21,22]. If a tank boils over or ground flowing fire occurs, the fire environment becomes very complex, and it is difficult to conduct direct and specific research, which is usually based on the research of thermal radiation. Therefore, this paper mainly focuses on the domino fire accident caused by thermal radiation.

At present, there are three methods to evaluate or calculate the probability of domino accidents caused by fire. One is damage threshold, which refers to the minimum intensity of the primary scenarios that can trigger the escalation. However, if the radiation generated by the fire exceeds the threshold of equipment damage, then the specific value of the actual damage probability cannot be obtained [23]. The second method uses CFD (Computational Fluid Dynamics, CFD) to establish a model to predict the fire behavior, and to then calculate the domino escalation probability. Omran Ahmadi et al., based on the experimental data of a 1 m crude oil pool fire and 30 m and 50 m diameter kerosene pool fires, built the consequence model for the tank area fire scenario. The FDS (Fire Dynamics Simulator, FDS) results showed that the studied dike pool fire might cause the domino effect in the tank area [24]. Pourkeramat et al. combined CFD and Abaqus, and studied the influence of wind speed, wind direction and soot concentration on the adjacent tank structure through radiant heat flux [25]. Scarponi et al. developed a 3D CFD model to analyze non-uniform fire scenarios, for example, LPG tanks exposed to partially engulfing pool fires [26,27]. The CFD method can retrieve the real-time fire scenario, but it requires detailed data as input, and requires a huge amount of calculation, which is difficult to achieve in traditional safety and risk assessment. Therefore, the third probability model method is the most commonly used method for quantifying the risk assessment of a domino accident caused by fire [28,29,30].

However, as a simplified model, the probability model has limitations in application [31]. In particular, after a fire escalation, it leads to fire in other oil tanks or facilities, which evolve into fire sources and have an impact on intact facilities. However, the traditional probability model does not take into account the synergistic effect of multiple fires occurring at the same time. As these models cannot take into account the change in the number of thermal radiation sources over time and the dynamic behavior of thermal radiation, they are greatly limited in evaluating the domino effect of third or higher levels [32].

The corresponding risk calculation mainly comes from different probability calculation methods. Wu et al. systematically studied the risk of oil tank fire caused by lightning, using traditional probability models [33]. Hou et al. proposed a risk quantification method for domino accident caused by double pool fire by using Bayesian network to calculate probability [34]. Sarvestani et al. also predicted the dynamic risk of propane tank by combining Bayesian network and fault tree analysis [35]. Based on the calculation of the total death probability of the probability model, Shou et al. proposed a dynamic individual risk assessment method for the simultaneous occurrence of toxic substance leakage and fire accidents [36].

Some studies began to make up for the above shortcomings. For example, Chen et al. proposed a method including the domino evolution graph (DEG) model and the minimum evolution time (MET) algorithm to simulate the evolution of domino accidents [37]. Zhou et al. defined a critical thermal dose for equipment failure, and improved the probability model of equipment failure on this basis, which could calculate the failure time of multiple fires when secondary fires occurred at different times [38]. Zhang et al., based on the adoption of the Agent-based Model and Simulation, modeled and simulated the spread of domino fire accidents, and considered the synergistic effect to realize analysis of domino accidents above the second level [39,40]. Ding et al. proposed a fire synergetic effect model (FSEM) to simulate the time–space synergetic effect of fire. Based on this model, a series of work was carried out, including the quantitative contribution of synergetic effect to domino effect risk [41], the visualization of domino accident risk [42], and the vulnerability of equipment under the coupling of fire and explosion [43]. With the continuous development of the probabilistic model method, the impact of time and space synergy on fire domino accident was considered, but the rapid calculation of equipment vulnerability parameters, as time changed, could not be realized.

Therefore, by combining the Monte Carlo algorithm and FSEM, we proposed a fast dynamic probability and quantitative risk calculation method for fire domino accidents in the storage tank area, considering the different thermal radiation effects of different failure order tanks on the target tank during the evolution process of the actual domino accident. We combined this with a case study for specific calculation and analysis. The case study found that the dynamic risk probability proposed in this paper could be used to calculate the domino accident of pool fire, and the failure probability of tanks B, C and D increased by 41.84%, 41.61% and 40.56% compared with that without considering the synergistic effect. An individual risk contour of 10^−5^/average year appeared.

## 2. Methodology

In this section, a new method, based on the Monte Carlo algorithm, to calculate the real-time probability and quantitative risk of tank domino accidents under the synergistic effect of pool fire, is explained in detail. As shown in Figure 1, the proposed method is divided into seven steps.

### 2.1. Collecting Basic Data

The first step is to collect necessary data for the new method proposed in this paper to calculate the real-time risk probability and quantitative risk of tank pool fire domino accidents. First, identify the tank area to be evaluated, including the number and spatial distribution of tanks. Secondly, collect the environmental conditions, such as temperature, humidity and wind speed. Finally, determine the initial parameters required for calculation (combustion heat of medium, tank wall thickness, tank wall temperature).

### 2.2. Calculating the Vulnerability Parameters of the Target Storage Tank

When a tank unit fire occurs, other tanks are subjected to the thermal action of the tank, which is called the target tank. The vulnerability parameters of the target tank are calculated, including the theoretical failure time of the target tank, the probability of accident escalation and the probability of failure changing with time.

According to the fire synergistic effect model (FSEM), proposed by Ding et al. [41], the thermal radiation intensity on the target unit is:(1)$$I=\sigma \cdot F_{view}{}_{,i}\cdot \alpha_{surface}\cdot \varepsilon_{fire}\cdot T{\left(t\right)}_{fire}^{4}$$
*σ* is the Stefan–Boltzmann constant, 5.67 × 10^−8^ W/(m^2^·K^4^); *F_view_*_,*i*_ is the view factor of the target tank unit relative to the failed tank *I*; *α_surface_* is the surface absorption rate of the target tank unit; *ε_fire_* is the emissivity of the storage tank of the failed fire source; *T*(*t*)*_fire_*_,*i*_ is the temperature of the failed fire storage tank *I* at time *t* (K).

In a fire accident, the equipment unit near the fire source is heated, due to the influence of thermal radiation, which leads to the failure, and then triggers the domino effect. The Probit model is often used to calculate the escalation probability of target storage tank unit under the effect of thermal radiation [44,45].
(2)$$EP=\frac{1}{\sqrt{2\pi }}\int_{-\infty }^{Y-5}e^{{{}^{-u}}^{{}^{2}}/2}du$$
(3)$$Y=12.54-1.847\ln\left(ttf\right)$$
(4)$$Pf=1-e^{-\frac{1}{ttf}t}$$
*EP* is accident escalation probability; *Y* is probability unit value; *Pf* is the probability of target tank unit failure with time; *t* is the time before failure(s).

For atmospheric tanks:(5)$$\ln\left(ttf\right)=-1.128\ln\left(I\right)-2.667\times 10^{-5}V+9.877$$

For non-pressure tanks:(6)$$\ln\left(ttf\right)=-0.947\ln\left(I\right)+8.835V^{0.032}$$
*ttf* is the theoretical failure time of the target tank unit (s); *V* is the storage tank unit volume (m^3^).

For the convenience of description, it is assumed that the number of tanks in the tank area is *m*, *i* is the failure sequence of the tank, the thermal radiation intensity of the target tank unit *j* is recorded as I*_i_*_,*j*_, the theoretical failure time is recorded as *ttf_i_*_,*j*_, the probability unit values are recorded as Y*_i_*_,*j*_, the accident escalation probability is recorded as *EP_i_*_,*j*_, and the probability that the failure of the target tank unit *j* with time is recorded as P*f_i_*_,*j*_. At this time, *I* = 1.

### 2.3. Determining the Next Failed Tank by Monte Carlo Simulation

Set simulation parameters, conduct Monte Carlo simulation for m − 1 target tanks respectively, and determine the earliest tank unit that fails. The Monte Carlo method, also known as the statistical simulation method, is based on the law of large numbers and uses the frequency of events as an approximation of the probability of events. The specific method involves using the Monte Carlo simulation algorithm, with the initial time t_1_ = 0, and the time step t = 1 s, to respectively generate a random number R*_j_*_,*k*_ for the target tank unit, the range of the random number is 0–1, and *k* is the number of iterations. If the value of R*_j_*_,*k*_ is less than P*f_i_*_,*j*_ (at this time, i = 1), it is considered that tank failure and accident escalation occurred. Find the tank unit T2 that failed the earliest and determine the simulated failure time t_2_. In this step, the stop time of a single simulation iteration is one of the most important parameters. In order to simplify the simulation, the fire intervention time, tint, is taken as this parameter [46]. According to previous studies, its value is 5 min in China [47].

### 2.4. Calculating the Equivalent Heat Radiation Flux of the Remaining Target Units

Regardless of the synergistic effect of time and space, in this case, the target tank unit is affected by tanks T1 and T2, and its thermal radiation intensity is the sum of all failure ignition sources. The vulnerability parameters of the remaining m − 2 target tanks are calculated respectively, including the thermal radiation intensity of the target tank unit I_eq,*j*_, and the comprehensive theoretical failure time *ttf*_eq,*j*_, equivalent heat radiation flux *Q*(*ttf*_eq,*j*_).
I_eq,*j*_ = I_1,*j*_ + I_2,*j*_(7)

According to the actual situation of the storage tanks, the comprehensive theoretical failure time *ttf*_eq,*j*_ of the remaining m − 2 target tanks can be obtained by substituting them into Equation (5) or (6).

The heat flux absorbed by the target tank unit can be calculated according to the following equation:(8)$$Q(ttf_{{}^{{}_{eq,j}}})=\int_{0}^{ttf_{{}^{{}_{eq,j}}}}q{\left(t\right)}_{absorbed,j}dt$$

Further, according to the Stefan–Boltzmann law:(9)$$q{\left(t\right)}_{absorbed,j}=\sigma \cdot \left(\sum_{i=1}^{n}F_{view}{}_{,i}\cdot \alpha_{surface}\cdot \varepsilon_{fire}\cdot T{\left(t\right)}_{fire}^{4}{}_{,i}-\varepsilon_{surface}\cdot T{\left(t\right)}_{{}_{surface,j}}^{4}\right)$$
*q*(*t*)*_absorbed_*_,*j*_ is heat flux absorbed by target tank unit *j* (W/m^2^); *n* is the number of fire sources; *ε_surface_* is the emissivity of the target tank unit surface; *T*(*t*)*_surface_*_,*j*_ is the surface temperature of the target tank unit *j* at time *t* (K).

The heating rate on the surface of the target tank unit can be calculated as:(10)$$\frac{dT{\left(t\right)}_{surface,j}}{dt}=\frac{q{\left(t\right)}_{absorbed,j}}{\rho \cdot c\cdot \delta }$$
*ρ* is the tank wall density of the target tank unit (kg/m^3^); *c* is the specific heat of the target tank unit wall, J/(kg·K); *δ* is tank wall thickness (m).

According to Equations (9) and (10), the comprehensive theoretical failure time *ttf*_eq,*j*_ is known, and the equivalent heat radiation flux *Q*(*ttf*_eq,*j*_) can be calculated.

### 2.5. Calculating the Vulnerability Parameters of Remaining Target Units under the Space–Time Synergistic Effect

Considering the synergistic effect of time and space, the vulnerability parameters of the remaining m − 2 target tank units, the synergy theoretical failure time *ttf*_co,*j*_, the probability unit value Y_co,*j*_, the accident escalation probability EP_co,*j*_, and the failure probability P*f*_co,*j*_ of target tank unit *j* with time can be calculated, respectively.

The specific approach is as follows: the synergy theoretical failure time *ttf*_co,*j*_ can be calculated according to the following formula:(11)$$Q(ttf_{{}^{{}_{eq,j}}})=\int_{0}^{t_{2}}q{\left(t\right)}_{absorbed,j}dt+\int_{t_{2}}^{t_{3}}q{\left(t\right)}_{absorbed,j}dt+\ldots +\int_{t_{n-1}}^{ttf_{co,j}}q{\left(t\right)}_{absorbed,j}dt$$

At this time, during the period from 0 to *t*_2_, the only heat source is the failed storage tank T1. From time *t*_2_ to the time when tank T3 fails, the heat sources are the failed tanks T1 and T2, and Equation (11) becomes:(12)$$Q(ttf_{{}^{{}_{eq,j}}})=\int_{0}^{t_{2}}q{\left(t\right)}_{absorbed,j}dt+\int_{t_{2}}^{ttf_{co,j}}q{\left(t\right)}_{absorbed,j}dt$$

According to Equations (11) and (12), given the equivalent heat radiation flux *Q*(*ttf*_eq,*j*_), the synergy theoretical failure time of the remaining m − 2 target tanks *ttf*_co,*j*_ can be obtained.

Further, according to Equations (2)–(4), the probability unit value Y_co,*j*_ and accident escalation probability EP_co,*j*_ of the remaining m − 2 target tank units are calculated, and the failure probability P*f*_co,*j*_ of target tank unit *j* with time is calculated.

### 2.6. Determining the Sequence of Failure Storage Tanks under the Space–Time Synergy Effect

The simulation parameters are set and the Monte Carlo simulation carried out on m − 2 target tank units, respectively, to determine the next tank to fail.

The specific method involves using the Monte Carlo simulation algorithm with initial time t = t_2_ and time step t = 1 s to respectively generate a random number R*_j_*_,*k*_ and *k* for the target storage tank unit. The range of random number was 0–1. Before the fire intervention time t_int_ = 300 s, if the value of R*_j_*_,*k*_ is less than Pf_co,*j*_, tank failure and accident escalation occur. It is necessary to find out the tank T3 with the earliest failure, and determine the simulation failure time t_3_.

### 2.7. Quantitative Area Risk Analysis

Repeat the above steps until the fire intervention time (t_int_ = 300 s) is reached, or all tanks have completed the calculation and simulation, so that the failure frequency of each tank can be obtained under the consideration of the synergistic effect, and then carry out quantitative risk assessment.

Quantitative Risk Assessment (QRA), also called Probabilistic Risk Analysis (PRA), is an important technical means for quantitative risk assessment. This method applies the principles and methods of safety system engineering, analyzes the failure probability and consequences of the system or equipment, and characterizes the risk as the product of accident frequency and accident consequences, so as to quantitatively describe the risk of hazard sources [48,49,50]. The core quantitative index of quantitative risk assessment is individual risk. Individual risk refers to the individual death probability of the person whose hazard source is generated at a fixed location, which is reflected as the risk contours [51,52]. The calculation process of individual risk is shown in Figure 2. The specific calculation formula and process are detailed in Appendix A. The quantification of human injury is usually calculated using the thermal radiation injury probability equation, which was improved by Pietersen on the basis of the empirical formula proposed by Buettner [53,54].

## 3. Case Study

To clearly demonstrate the application process of the proposed method, and to verify the applicability of the proposed method a case study was conducted. A tank group, containing four atmospheric tanks, was selected as the research object. The plane layout is shown in Figure 3. The selected tank group is the position marked with a red box in the figure. The size of each tank is the same. For the convenience of narration, the tanks are numbered as A, B, C and D respectively. The diameter and height of the tank were 46,000 mm and 19,580 mm respectively, and the volume was 30,000 m^3^. The stored material was gasoline. The parameters ε_fire_ = 0.7, ε_surface_ = 0.7, α_surface_ = 0.7, T_fire_ = 1177 K, other input parameters, were set according to reference [41].

In order to simplify the research and calculation, the following basic assumptions were made:It was assumed that tank A caught fire first;Only thermal radiation was considered, and only pool fire was assumed as an accident scenario;Regardless of the development time of the fire, the temperature of the fire was considered to be average.

## 4. Results and Discussion

According to the above invented algorithms and calculation examples, Python programming was used for calculation, and the fire rescue time was set as 300 s. Assuming that storage tank A began to fail first, 10,000 iterations were carried out to obtain the calculation result.

The first failed tank was denoted as T1, and the subsequent failed tanks were denoted as T2, T3, and T4 according to the time of occurrence. According to the number of failed tanks and failure sequence, there were 16 accident scenarios without considering that different tanks would fail at the same time. The tank failure sequence corresponding to each accident scenario is shown in Table 1.

The accident scenario results obtained by simulation were counted. The occurrence times of each accident scenario are shown in Figure 4.

It was not necessary to consider the synergistic effect and it was assumed that tank A failed first. According to Equations (2)–(6), the theoretical failure times of tanks B, C and D were 327 s, 327 s and 390 s, respectively. The curve of tank failure probability with time is shown in Figure 5 below.

It can be seen from Figure 5 that the comprehensive real-time failure probability curves of tanks B and C were the same, while the fluctuation of the comprehensive real-time failure probability curves of tank D was relatively small, that is, tank D was the least likely to fail first under the action of thermal radiation of tank A. If tank B failed first under the action of thermal radiation from tank A, the statistics of the occurrence times of tank B failure time under this working condition could be obtained according to the Monte Carlo simulation, as shown in Figure 6. The respective synergy theoretical failure time *ttf*_co,C_ and *ttf*_co,D_ of storage tanks C and D were calculated, and the curve of *ttf*_co,C_ and *ttf*_co,D_ varying with the failure time of storage tank B was obtained, as shown in Figure 7. It can be seen from Figure 7 that failed tanks A and B had similar synergistic effects on tanks C and D. The earlier tank B failed, the smaller the synergistic theoretical failure time of tanks C and D, that is, the more likely they were to fail. Due to the fire intervention at 300 s, if tank B failed after 250 s, tank D would not fail, while tank C would not fail until the failure time of tank B reached 290 s.

The failure times of tank B in the simulation results were selected as 20 s, 55 s, 105 s and 219 s. At this time, the synergetic theoretical failure time of tank C and tank D was calculated, as shown in Table 2. The corresponding synergetic theoretical failure probability *Pf*_co,C_ and *Pf*_co,D_, respectively, were calculated, as shown in Figure 8 and Figure 9. By comparison, it can be seen that the later the failure time of tank B was, the smoother the real-time failure probability curve of the remaining tank was.

If tank C failed first under the synergistic thermal effect of tank A and tank B, the successive failure times of tanks B and C, under this working condition, could be obtained, according to the Monte Carlo simulation results, so as to calculate the theoretical failure time *ttf*_co,D_ of tank D under the synergistic effect of the three failed tanks, as shown in Figure 10.

In the simulation results, 6 different theoretical failure times of tank D under the synergistic effect were selected, as shown in Table 3. The synergetic theoretical failure probability P*f*_co,D_, corresponding to tank D, was calculated, as shown in Figure 11. It can be seen that the smaller the successive failure time of tank B and tank C, the steeper the real-time failure probability curve of tank D, that is, the higher the failure probability under the same time.

The calculation of accident consequence and individual risk in this paper was carried out by using Quantitative Risk Assessment software for major hazard installations software, CASSTQRA (V2.0, China Academy of Safety Science and Technology, Beijing, China). We inputted the information of field investigation, analysis and sorting into the regional quantitative risk assessment software system, and the calculation of individual risk, tracking and drawing of contours could be automatically completed [55].

The basic leakage probability adopted the reference value in the method for determining the external safety protection distance of hazardous chemical production devices and storage facilities (GB/T 37243-2019) [56], and the failure probability caused by the synergistic effect was calculated by Equations (13) and (14), as shown in Table 4.

CASSTQRA V2.0 can be used to obtain the casualty radius of each tank in case of pool fire. The red line is the radius of death, the blue line is the radius of serious injury, and the green line is the radius of minor injury, which were 10,347 mm, 12,190 mm, and 16,962 mm, respectively, as shown in Figure 12.

In combination with Table 4, the failure probability in CASSTQRA V2.0 was corrected, and the individual risk contour was used to represent the individual death probability of the personnel at the fixed position, as shown in Figure 13. Among them, the individual risk values represented by red, pink, orange and yellow lines were, respectively, 1 × 10^−5^, 1 × 10^−6^, 1 × 10^−7^, 1 × 10^−8^/average year. Figure 13a,b indicate the individual risk of tank fire accident without considering the synergistic effect and considering it, respectively. It can be seen that when considering the synergistic effect, the individual risk value increased greatly, and an individual risk contour of 10^−5^/average year occurred.

## 5. Conclusions

In this study, a new risk probability and quantitative risk calculation method is proposed to solve the problem that different failure order tanks produce different thermal radiation to target tanks in the evolution process of a fire domino accident in the actual storage tank area. The equivalent heat radiation flux is obtained according to the incident heat radiation intensity received by the target unit, and the synergy theoretical failure time is calculated by the equivalent heat radiation flux. The time parameter is used as the judgment criterion, all accident scenarios are explained by the Monte Carlo algorithm, and the correction value of failure probability is obtained.

This method can not only understand the evolution of multiple escalation fire domino accidents, when considering the synergy effect, but can also obtain all possible accident scenarios and corresponding vulnerability parameters, including synergy theoretical failure time, accident escalation probability EP and target unit real-time failure probability P*f.* The case study took four storage tanks as the research object, showed the failure risk probability and individual risk of different storage tanks under the synergy effect, and illustrated the effectiveness and superiority of the proposed method.

This study is the first to model the spatiotemporal evolution of domino effects using Monte Carlo and FSEM. The research results can be used to support the extension of risk assessment of key facilities in chemical industrial parks and the decision of emergency rescue. However, this study only considered the escalation of thermal radiation caused by pool fire. In actual chemical accidents, the escalation of explosion overpressure, explosion debris, and even the superposition damage caused by the leakage and diffusion of toxic and harmful gases, may also occur. Therefore, this work needs to be further improved.

## Figures and Tables

**Figure 1 ijerph-19-16483-f001:**
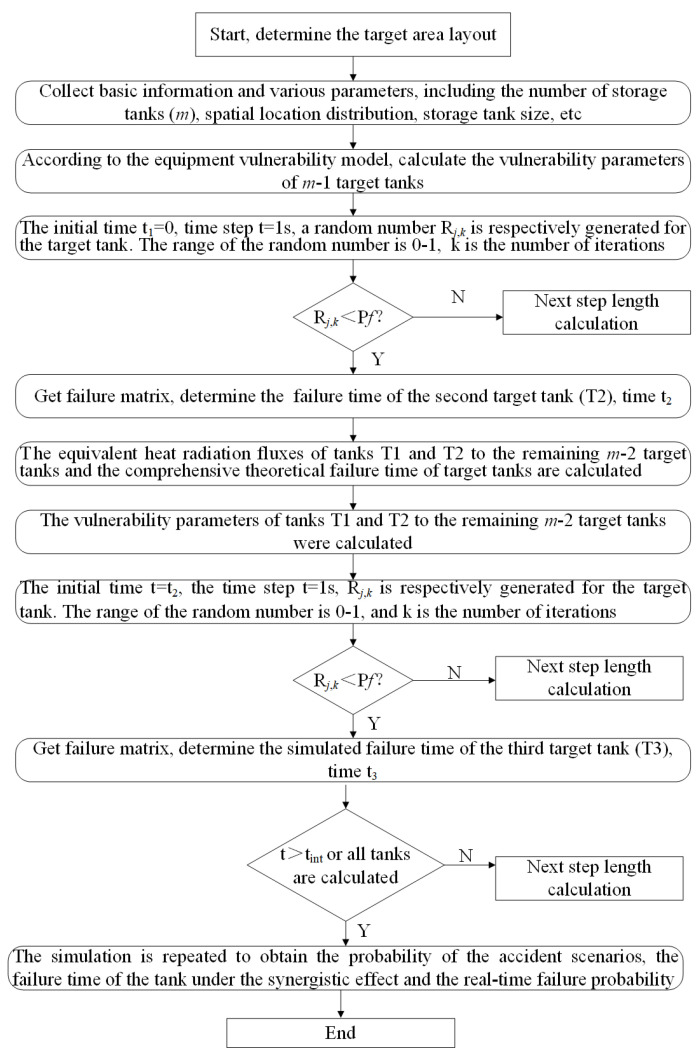
The flowchart of the method based on the Monte Carlo algorithm.

**Figure 2 ijerph-19-16483-f002:**
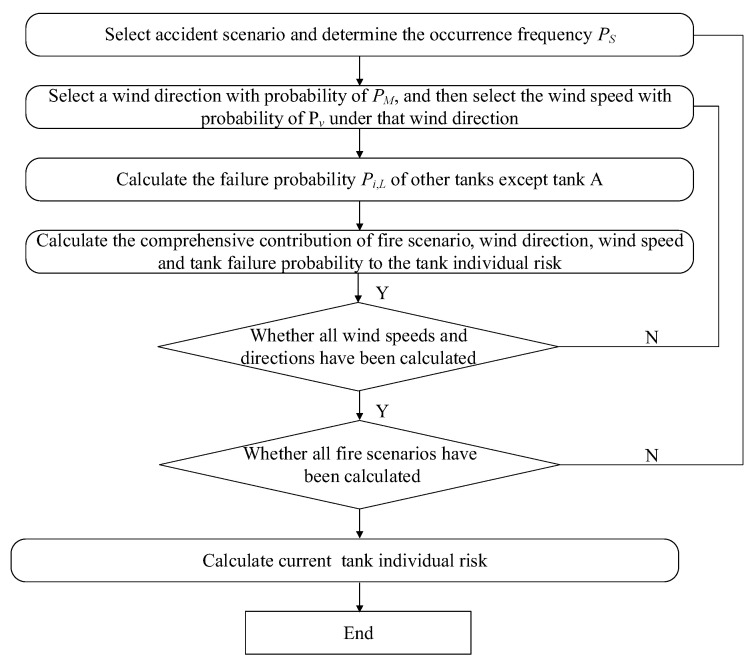
Tank failure risk calculation program.

**Figure 3 ijerph-19-16483-f003:**
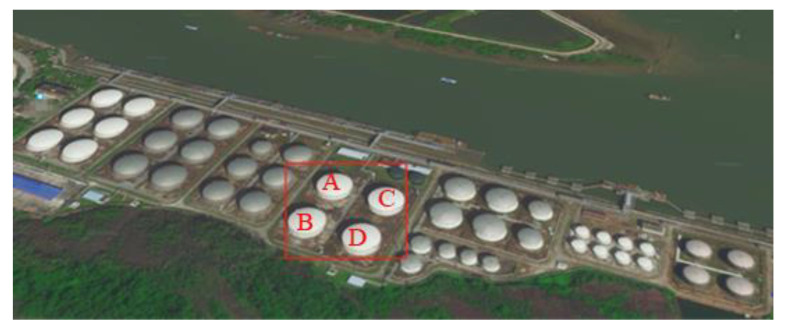
Layout of tank area.

**Figure 4 ijerph-19-16483-f004:**
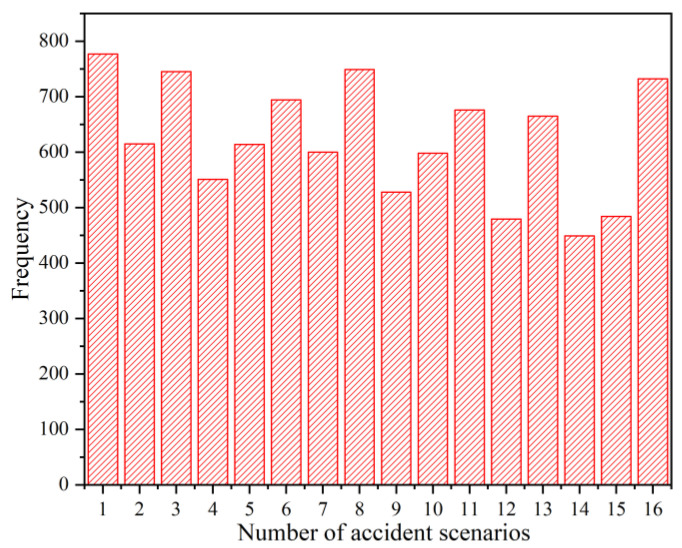
Type statistics for accident scenarios.

**Figure 5 ijerph-19-16483-f005:**
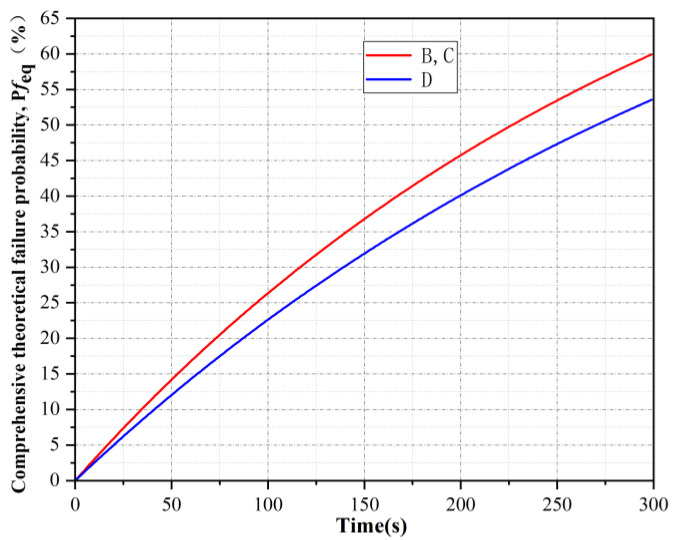
Comprehensive real-time failure probability of remaining tanks after tank A fails.

**Figure 6 ijerph-19-16483-f006:**
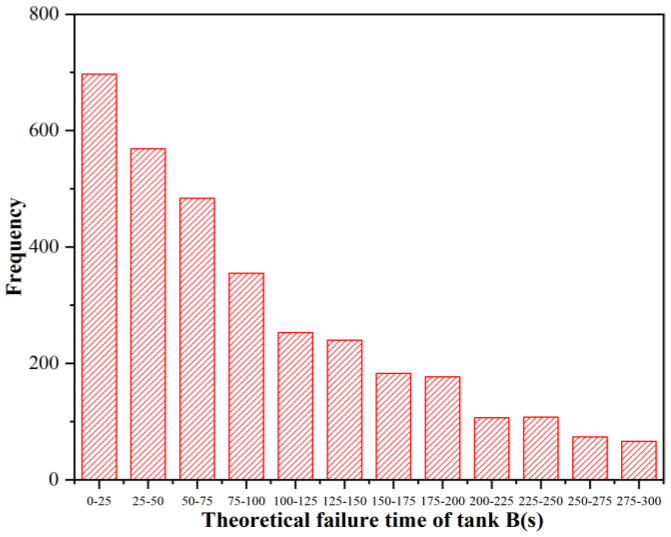
Theoretical failure time of tank B.

**Figure 7 ijerph-19-16483-f007:**
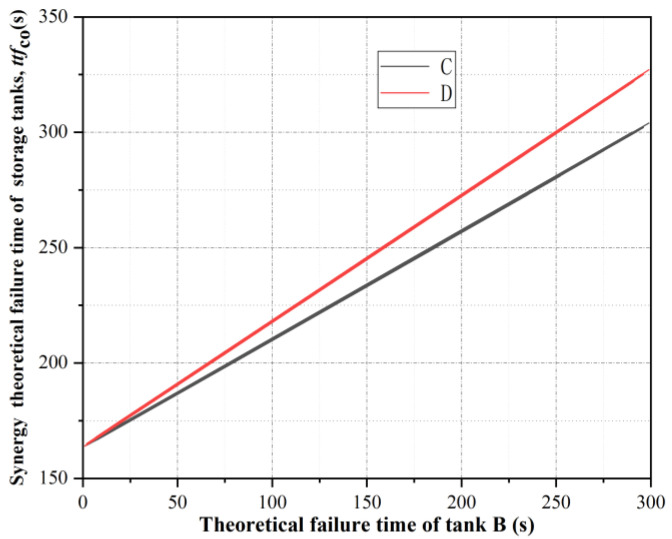
Synergy theoretical failure time of tanks C and D.

**Figure 8 ijerph-19-16483-f008:**
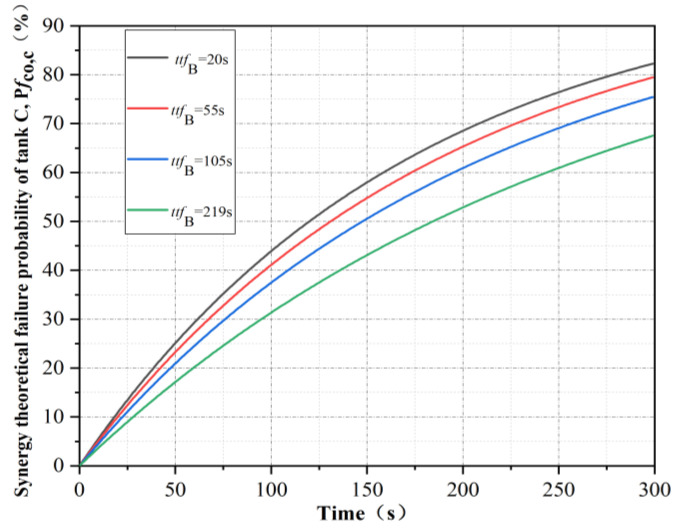
Real-time failure probability of tank C under the synergistic effect of tanks A and B.

**Figure 9 ijerph-19-16483-f009:**
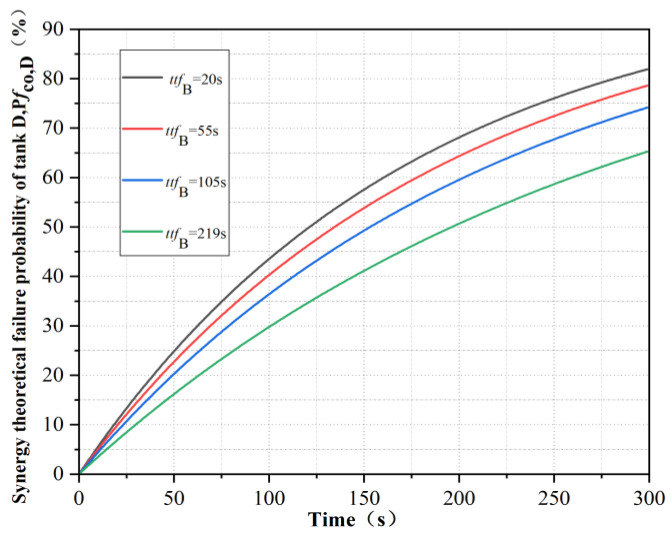
Real-time failure probability of tank D under the synergistic effect of tanks A and B.

**Figure 10 ijerph-19-16483-f010:**
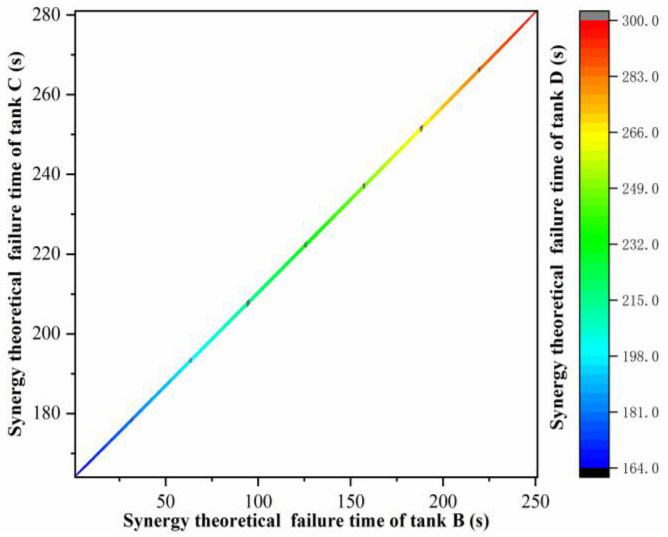
Theoretical failure time of tank D under the synergistic effect of tanks A, B and C.

**Figure 11 ijerph-19-16483-f011:**
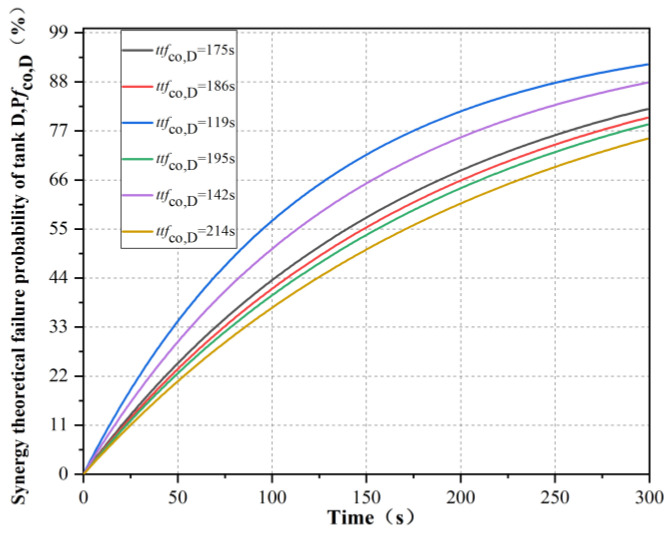
Real-time failure probability of tank D under the synergistic effect of tanks A, B and C.

**Figure 12 ijerph-19-16483-f012:**
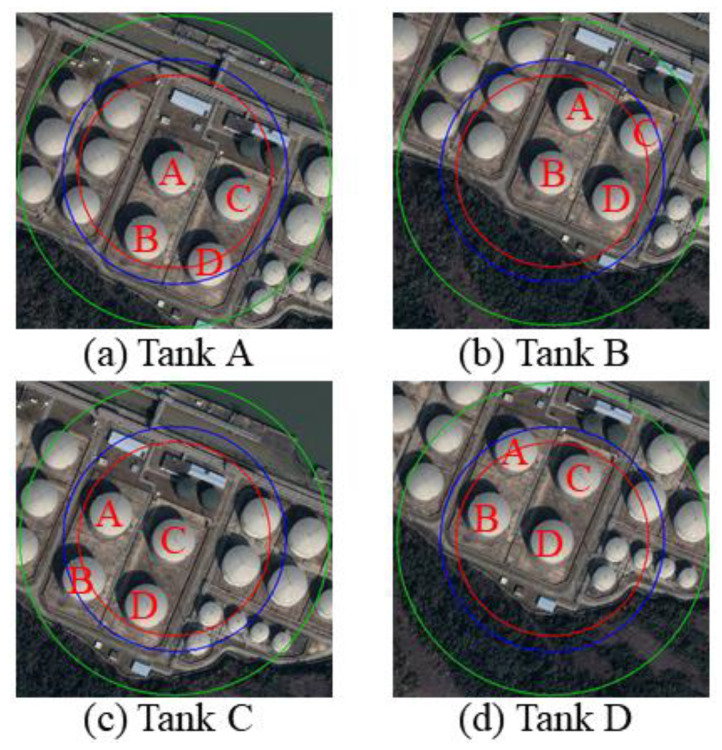
Casualty radius of tank pool fire.

**Figure 13 ijerph-19-16483-f013:**
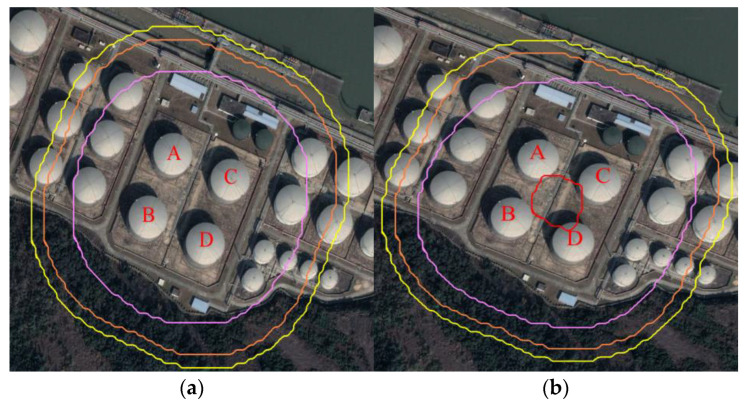
Individual risk analysis of storage tanks. (**a**) Without considering the synergistic effect (**b**) Considering the synergistic effect.

**Table 1 ijerph-19-16483-t001:** Tank failure sequence.

Number	T1	T2	T3	T4	Number	T1	T2	T3	T4
1	A	B	C	D	9	A	C	D	
2	A	B	C		10	A	C		
3	A	B	D	C	11	A	D	B	C
4	A	B	D		12	A	D	B	
5	A	B			13	A	D	C	B
6	A	C	B	D	14	A	D	C	
7	A	C	B		15	A	D		
8	A	C	D	B	16	A			

**Table 2 ijerph-19-16483-t002:** Synergy theoretical failure time of tank C and D.

Failure Time of Tank B (s)	Synergy Theoretical Failure Time of Tank C (*ttf*_co,C_)	Synergy Theoretical Failure Time of Tank D (*ttf*_co,D_)
20	173	175
55	189	194
105	213	221
219	266	283

**Table 3 ijerph-19-16483-t003:** Theoretical failure time of tank B, C, D under synergy effect.

Failure Time of Tank B (s)	Synergy Theoretical Failure Time of Tank C (*ttf*_co,C_)	Synergy Theoretical Failure Time of Tank D (*ttf*_co,D_)
20	31	119
	62	195
55	62	142
	214	214
105	107	175
	138	186

**Table 4 ijerph-19-16483-t004:** Failure probability of storage tank.

Tank	Failure Mode	Type of Accident	Base Leakage Frequency	Failure Probability Due to Synergetic Effect	Total Failure Probability
A	Break up completely	Pool fire	0.00002	0	0.00002
B			0.00002	0.00001439	0.00003439
C			0.00002	0.00001425	0.00003425
D			0.00002	0.00001365	0.00003365

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
