# Peer review of "Study on Dynamic Probability and Quantitative Risk Calculation Method of Domino Accident in Pool Fire in Chemical Storage Tank Area"

_ijerph, 2022, doi:10.3390/ijerph192416483_

Round 1

Reviewer 1 Report

I have reviewed the paper entitled "Study on dynamic probability and quantitative risk calculation method of domino accident in pool fire in chemical storage tank area". This is a useful study, quantitative risk calculation of tank fire is still an important problem, and a novel method is proposed in this manuscript. Therefore, the paper can be accepted for the publication after some revisions.

* This paper focuses on the probability and risk calculation of tank pool fire, but the literature review summarizes the development and classification of probability calculation as well as the shortcomings, so the literature on quantitative risk calculation needs to be supplemented in the literature review.

* This paper uses Monte Carlo to simulate the failure sequence and scenario of the tank. What is the principle of this algorithm? It needs to be supplemented.

* In the case analysis, the size and volume of the tank and the type of substances stored are given in this paper, but this is far from enough for the quantitative calculation of risk, so it is necessary to further clarify the parameters of the given case.

* What is the basis used to determine the failure of the tank?

*Authors need to carefully check the details in the article and make corrections.

Reviewer 2 Report

Manuscript number: ijerph-2080214

Title: Study on dynamic probability and quantitative risk calculation method of domino accident in pool fire in chemical storage tank area General Comments.

The article explains a combination of the Monte Carlo algorithm and FSEM techniques and proposed a fast real-time probability method in a particular fire domino accident that might occur in the storage tank area. The manuscript is well written however, a few points are essential to consider before its final publication.

The following suggestions are given for improvement in the manuscript.

 Specific Comments:

1)      Abstract:  The abstract only contains general information. It is suggested to add some background with a few objectives, and possible applications of this study and highlight the novelty of this work.

2)      In the introduction section, the last paragraph must elaborate on the key findings of this study.

3)      Section 2: Please improve the quality of Figure 1.

4)      Please use the same units in line 275.

5)    Figures 3,12 and 13: Please specify all tanks with their exact assigned names.

6)      Modify and remake Figure 10 with better visualization because it is challenging to read the axis description.

7)     Finally, the manuscript seems very lengthy and it contained so much information in the material and method section. So I can suggest that you can move some of the data to a separate supplementary data file. Copy some tables and equations in that file.
